# Lessons in Stories: Why Narrative Medicine Has a Role in Pediatric Palliative Care Training

**DOI:** 10.3390/children8050321

**Published:** 2021-04-22

**Authors:** Natalie Lanocha

**Affiliations:** Department of Pediatrics, Oregon Health and Science University, Portland, OR 97239, USA; lanocha@ohsu.edu

**Keywords:** narrative medicine, reflection, pediatric palliative care

## Abstract

Narrative medicine is introduced and explored as a potential tool for developing competency in medical training, including reduction of burnout, sustaining empathy, and allowing for reflective practice. Developing cultural humility, communication skills, ethics, community building, and advocacy are also reviewed as domains that may be bolstered by training in narrative. Applications specific to pediatric palliative care are suggested, along with avenues for further research.

“Baby R, 6 month old male with fever and altered mental status” flashed across my pager sometime after midnight on my last shift as a resident in the Pediatric Intensive Care Unit. Peering into the crib of this urgent admission, I noted a sleepy infant with porcelain skin and limbs hanging limp like a ragdoll. The occasional whimper conferred a feeble sign of life. His parents nervously swayed around the crib in an attempt to both stay out of the way of the care team and cling as close to his side as possible. When he was settled with monitors attached, the story poured out from his mother. The details were exquisite: an accurate history was one of the family’s limited tools to aid in their son’s ICU transfer. I learned that baby R developed a high fever and lethargy several days ago. He stopped feeding and had not offered a smile since the morning. As antibiotics continued their course through his IV line, lumbar puncture results returned from the emergency room confirming meningitis. My tone confident and hopeful, I reassured the family that he was getting the medical care he needed and they had done the right thing to bring him in. I explained that often children spend weeks in the hospital with meningitis, but typically they do improve.

The night was quiet, and I sat down to get to know baby R and his family. R was their first child, and it had taken some time to get pregnant. He recently started daycare, his first separation and a test of his parents’ anxiety. He loved books and being held by his mom. His dad took out a phone and began flipping through photos of a smiling baby, earnestly sharing evidence of his son’s previous health. I looked again at the pale, floppy form in the crib and stifled a wince.

My memories shuffle a bit from there, but they remain crushingly painful. Baby R developed nystagmus, his stormy gray eyes darting back and forth. His blood pressure climbed and his heart rate plummeted. A breathing tube was deftly inserted into his airway. Shortly after, his eyes stopped their dizzying rotation, and his pupils became wide and fixed. A repeat image of his brain, not six hours from the last, showed his brainstem herniated into the bony ring that formed the base of his skull. He had developed one of the worst complications imaginable from meningitis. My twenty-eight-hour shift over, I shakily brought his parents a cup of coffee on the worst morning of their lives.

Not for the first time, I left the hospital with sobs wracking my body, doubled over in pain. I felt guilty that I had offered reassurance, and guilty for my profound despair after, which I knew paled in comparison to what baby R’s parents were experiencing. When I returned to the hospital the next day, R had died, his parents long gone. I felt lost, confused, and temporarily crippled with grief. I wondered, how do I return to clinical duties and work as an effective pediatrician?

## 1. A Need for Narrative

Over time I have discovered that emotionally challenging encounters, such as my experience with baby R, are intrinsic to clinical medicine, but it is our response to these events that impacts how we can productively move forward and avoid burnout. Now, as a pediatric palliative care (PPC) fellow, I find myself intentionally searching for tools to process emotional interactions and build resiliency. Burnout, the phenomenon of mental and physical exhaustion associated with work or caregiving stress, is unfortunately commonplace in medical education [1]. Recently, in the Common Program Requirements, the Accreditation Council for Graduate Medical Education (ACGME) acknowledged the high risk of burnout in trainees. In a statement, the Council writes, “Residents and faculty members are at risk for burnout and depression. Programs, in partnership with their Sponsoring Institutions, have the same responsibility to address well-being as other aspects of resident competence” [2]. With this acknowledgement, the ACGME emphasizes the importance of building skills for sustainable self-care, and charges graduate medical programs to add resiliency building to their medical education agendas.

PPC providers are uniquely exposed to repeated instances of children suffering from serious illness and loved ones navigating tragedy. In a survey of PPC physicians, distress around a clinical situation, physical exhaustion, and recent personal loss were all shown to be significant determinants of compassion fatigue, a secondary traumatic distress following contact with patients’ suffering [3]. In a field where loss is ubiquitous and grief is witnessed almost daily, it is crucial to bolster protective factors and nurture job-related fulfillment. Narrative medicine is one such tool that provides a framework for processing complex emotion and promotes the development of sustainable methods for self-reflection. As I will describe, narrative competency can be utilized in medical training and practice to sustain empathy, promote resiliency, and build skills for delivery of excellent palliative care. We may consistently and reliably turn to narratives, even in our darkest moments, in order to take care of ourselves and provide the best care for our patients and families.

## 2. Developing Narrative Competence

At its core, narrative medicine entails a profound respect for patient stories. This tenet of effective medical practice can be found woven into centuries of medical teaching. Sir William Osler embraced this notion, maintaining that “the good physician treats the disease; the great physician treats the patient who has the disease.” Rita Charon, a medical internist, narrative medicine pioneer, and founder of the Program in Narrative Medicine at Columbia University, has taken Osler’s charge a step further. Charon suggests, “along with scientific ability, physicians need the ability to listen to the narratives of the patient, grasp and honor their meanings, and be moved to act on the patient’s behalf” [4]. Accomplishing this starts with careful listening. Before we act, we must stop, witness, and learn, an idea central to the principle of “first, do no harm”. In order to train ourselves in careful listening and the ability to grasp narrative, we can turn to the humanities. By studying stories in addition to biology, medical trainees may become familiar with literary methods, with the goal of attuning ourselves to patients’ narratives to provide whole-person care.

In contrast, postgraduate training often forces trainees to condense patient stories for the sake of efficiency. Trainees are encouraged to utilize the “one-liner”, such as the sign out I received for baby R, which serves to reduce a complex history into a single sentence to facilitate swift rounding and transitions in care. In addition to abbreviating, this can have the effect of desensitizing the clinician by distancing providers from the full narrative [5]. Instead of condensing, narrative medicine offers the opportunity for expansion, with the goal to improve understanding of patients’ lived experiences. By attending to the full story and any emotions revealed in storytelling, we are provided a clearer window into our patients’ lives. It may be tempting to summarize clinical scenarios in order to create shortcuts or even shield ourselves from deeper involvement that can potentially lead to grief. Still, the full narrative allows greater insight into patients, encourages humanization, and even helps providers to begin the process of healing from our work.

## 3. Perception and Representation

Practically speaking, developing proficiency in narrative competence involves immersing oneself in both reading and writing. In reading, we learn to perceive events and persons fully, while writing offers the opportunity to capture these perceptions through representation [6]. When reading texts closely, we are asked to consider not only the content of the story, but also its form (temporal course, silences, subplots), and performance (methods of telling) [7]. Through close study of literature, we can train our focus on these aspects of verbal and nonverbal communication. Once learned, the skills can be extrapolated to use in patient encounters. With greater attention to silences, gestures, and expressions, along with the history offered, providers can more accurately perceive a patient or family’s experience, including their hopes, fears, and understanding of an illness.

It is only once we have truly perceived that can we attempt to represent. Representation through writing involves capturing reality and making it accessible to others [6]. Through representation, perceptions take material form. Accomplishing this requires effective comprehension of the moment, combined with the communication skills to characterize it. Practicing representation may be achieved through writing composition. Similar to perception, once honed, these skills are transferrable to patient care. In clinical care, perception and representation come together through the acts of bearing witness, compassionately acknowledging, and subsequently representing a patient’s suffering through written documentation and communication. These are requisite doctoring skills, and they are central to palliative care.

## 4. Reading to Promote Empathy

While the inclusion of literature in medical education may seem like a non-traditional tool, its use can help with establishing proficiency in perception and developing and sustaining empathy. Theory of Mind describes the ability to understand others’ perspectives, allowing an individual to “walk in another person’s shoes” [8]. In reading literature, we delve into characters’ subjective experiences. Still, the inherent nature of fiction provides a layer of protection: we can practice seeing from a character’s point of view without risking potential consequences of real-world attachments. In a series of elegant experiments, researchers Kidd and Costano demonstrated that reading literary fictions can enhance Theory of Mind, and may help clinicians find lasting empathy in clinical scenarios [9]. Applied in a medical education context, exploring literary works, particularly in a group setting, provides a safe environment for trainees to practice empathy together.

This makes good sense in theory, but in practice reading literature has also been shown to have a direct impact in decreasing burnout in palliative care physicians, and potentially in bolstering empathy. To assess the problem of burnout, Marchalik et al. used the two-item Maslach Burnout Inventory [10] in a survey of hospice and palliative medicine (HPM) providers. The team discovered that in the population of HPM survey respondents, 16.6% met the criteria for high emotional exhaustion (“I feel burned out from my work”) and 6.3% met the criteria for high depersonalization (“I have become more callous toward people since I took this job”) [11]. In the same study, it was shown that providers without burnout were more likely to be consistent readers of non-medical literature, thus suggesting that engaging with literature of this kind may actually be a protective factor against burnout in HPM providers [11]. Given that the main protective effect was observed across the domain of depersonalization, the authors concluded that reading fiction may help to encourage empathy by strengthening Theory of Mind, thereby reducing provider callousness toward patients. Thus, within the palliative care provider population, reading narrative may be considered in the work to prevent burnout and bolster positive empathy.

## 5. Writing for Communication and Reflection

Writing in medicine comes in many different forms, and can be utilized to help with effective representation, communication, and reflection. Honing writing skills has obvious benefits for clinicians across the care spectrum, including developing clear and effective note writing. As providers, we are accustomed to writing daily notes in order to communicate our findings. While these daily notes have historically been tailored to communicate among physicians, they are now becoming increasingly open to patients, and may be a tool to aid in the development of a mutually beneficial physician-patient partnership [12]. A well-written note that provides a clearly articulated understanding of the patient’s suffering with an appropriate plan may demonstrate comprehension on behalf of the physician as well as expression of compassion. In this case, it is possible the note could even have a therapeutic benefit for the patient [13].

Yet, beyond our daily writing, intentional reflective writing may be a tool in response to emotionally challenging experiences. Although “reflection” has multiple connotations, it is helpful to utilize a framework to understand how this practice can be applied in the context of medical education. In a model by Wear et al., there are four major elements of reflection: interrogating or elaborating on an experience, “working out” an issue arising from an experience, processing an experience where there is no obvious solution, and ultimately, pursuing transformative action as result of the reflection [14]. As envisioned here, reflection is a complex and active process that requires significant work and self-awareness.

When considering reflection in this manner, writing emerges as an ideal device to facilitate the reflective practice. Writing involves multiple iterations or drafts, and can be seen as a dynamic process. Before we even begin, we are required to pause and consider. Writing that involves “working out novel content” can be understood as *knowledge transforming*, which contrasts purely descriptive writing (*knowledge telling*) [15]. We may begin by restating the facts, but ultimately, as we search for words, we weave in our own subjective perspective. Knowledge transforming writing demands personal reflection, including examination of and dialogue with our past, present and future selves [14]. As we write, we search within ourselves, asking, what emotions or memories are triggered by scrutinizing a certain experience or patient encounter? The act of writing thus allows a return to these emotions for additional processing.

Becoming aware of one’s own feelings associated with a particular patient scenario often prompts a greater empathetic response toward the patient. Classically, when I consider my own sorrow in sharing a diagnosis of cancer with a patient, my empathy for the patient and family intensifies. In turn, when looking back on a scenario and appreciating personal feelings of generosity or nurturing directed toward a patient, clinicians may be prompted to hold this same level of care for themselves [16]. For medical trainees, reflective writing can prompt a search for meaning in difficult interactions, and may spark a change in practice to better serve patients and care for the self, allowing for continuous growth and evolution of the physician [17]. Following narrative writing curricula, trainees themselves have noted deeper self-reflection, enhanced self-awareness, and stronger motivation to improve [18]. These skills are crucial in cultivating empathy, building resilience, and fostering growth.

When it comes to unpacking intense emotional experiences and trauma, writing can be utilized specifically as a therapeutic tool. In a classic psychology study, Pennebaker et al. demonstrated that six weeks after writing about past trauma, otherwise healthy undergraduates had increased positive mood reports, fewer illnesses, increased cellular markers of immune functioning, and fewer visits to the student health center in comparison to matched controls [19]. In a more modern clinical setting, narrative writing with emphasis on affirmation, legacy making, and mindfulness was also shown to increase resilience and decrease depressive symptoms and perceived stress [20]. When used with trainees, early inquiries show that reflective writing may have a similar healing effect. First year internal medicine interns described catharsis through the therapeutic act of writing, and in particular noted that detailing previous “bad” experiences provided some emotional release [18]. It seems that utilizing narrative writing has great promise as a tool for building resiliency in both budding and established physicians.

## 6. Utilizing Narrative to Develop Doctoring Skills

Medical training seeks to bolster skills deemed essential for “good” doctoring. Developing skills in physical exam, diagnosis, clinical reasoning, and prognostication are all essential; however, a well-rounded and trustworthy physician is expected to have additional competencies that traditional medical training may only touch upon. Narrative medicine curricula can provide an effective supplement to hone many of these skills that are critical to good doctoring and essential to provision of palliative care, can help to foster community, and ultimately, can move physicians to honor and advocate for their patients (Figure 1).

Exposure to narrative improves our ability to provide quality care for diverse populations. Establishing “cultural humility” as defined by Tervalon and Murray-García involves commitment to continuous self-reflection, readdressing power differentials, and developing advocacy partnerships, and has been proposed as one way to go about limiting outright and hidden biases in medicine [21]. As described, attaining cultural humility requires introspection in addition to learning skills for provision of individualized and just care for each patient. By eschewing cultural stereotypes and characterizing individual identities—which may be quite heterogenous even within a single depicted cultural group—specific literary texts can help to encourage examination of closely held biases. Moreover, when used with medical trainees, these works may prompt reflection and promote discussion in a safe and open setting [22]. In a Literature and Medicine course for medical students, exposure to fiction was seen to stimulate analysis of personal belief systems and encourage students to challenge their preconceived notions [23]. Opening oneself to narrative studies may thus supplement other skills learned in unconscious bias training, anti-racism workshops, and other educational endeavors to promote provision of just care to all patients.

Another important skill for palliative care providers is the ability to tolerate and manage uncertainty. While prognosis itself has inherent uncertainty, additional consideration of our patients’ values means that there is often more than one right answer when it comes to care decisions [24]. In pediatric oncology, directly addressing uncertainty has been noted to allow space for parents to share worries early on, and with a clinician with whom the family has come to know and respect [25]. Thus, there is a pressure for providers to learn to identify uncertainty, address it with families, and tolerate it themselves. Engaging with narratives offers an opportunity to practice these skills. As readers, we eagerly anticipate the telling of a story, and for a time, we are challenged to tolerate uncertainty as the story unfolds [7]. As we look for resolution in a narrative, we may consider and plan for possible outcomes, encouraging the practice of critical palliative medicine skills within the safe space of fiction.

Giving close attention to stories can also enhance skills in communication. In their thoughtful curricular design, narrative medicine programs can intentionally utilize certain texts to draw out receptive and expressive communication skills. Upon completion of the Columbia University narrative medicine elective, a class that includes poetry and other literature, students described development of skills including self-awareness, articulation, observation, patience, among others [26]. Similarly, Taiwanese medical students who completed a narrative medicine course were found to perform higher than their peers without the narrative training during testing at communication stations within an objective structured clinical exam (OSCE) [27]. This type of training has potential applications specific to palliative care as well. In a narrative training directed toward interdisciplinary palliative care providers in the UK, exposure to storytelling was felt to help facilitate better interprofessional communication [28]. It seems that more study could be directed toward learning about the intentional use of narrative for communication training, however, the research thus far is very promising.

Ethics is another area that benefits from close attention to narrative. Charon argues that narrative competence can help clinicians to recognize the ethical problem and then put the problem into words, something that must be done effectively prior to debating the issue [29]. In the clinical context, this begins with close listening of patient stories to understand values, cultural context, and history of past experiences with the medical system. Ethical deliberation can only truly begin once full understanding of the problem and the various viewpoints of those concerned is realized. It is even possible that in elucidating the problem, a solution becomes more apparent.

Narrative medicine practice also serves to foster a sense of community. In a story-telling venue, medical faculty and staff were invited to share stories from the hospital, and in doing so learned that they were “not alone” in their experiences [30]. In contrast to much of classical medical training, hierarchy is limited in the study of narrative. For many, there is a sense of vulnerability in leaving the medical textbooks to open a novel. Yet, when we come together to discuss literature, we face this vulnerability as a collective group. In addition, by engaging in non-traditional texts, bridges can be built to other scholarly departments. Encouragingly, narrative training appears to benefit from participation across disciplines when programs include scholars in the arts, music, and literature [17]. Through narrative medicine, participants may build larger networks to encourage the development of creative, competent, and supported providers.

Beyond helping clinicians grow, narrative medicine curricula may also help providers to dignify patients suffering from serious illness and promote advocacy on their behalf. In her work, Charon brings up with some frequency writing as a means of honoring the stories of illness [4,6,21]. Through perception, we absorb and interpret our patient’s suffering. Yet, the work does not stop there. When we are attuned to patients with a narrative lens, we provide companionship in challenging times. As we move to representation, we may then reveal our comprehension of patient suffering and honor their story. To complete our duty as narratively competent physicians, we then allow the stories to move us to act. We thoughtfully listen, permit stories to change our viewpoints, and become motivated to make changes on the behalf of our patients. It is here that advocacy intersects with narrative medicine, but it is on us as providers to take the next steps for transformative action as we truly commit to our patients’ wellbeing.

## 7. Tailoring Narrative Medicine Programs and Research to Our Needs

Use of narrative in medicine can aid in our teaching of necessary skills for doctors. However, taking the next step in developing valuable programmatic content can feel daunting, and comes with certain challenges. While a variety of studies look into the use of reading and expressive writing to inspire growth in medical trainees, the content of the programs is somewhat variable [31]. A recent systematic review of narrative medicine programs shows that most programs take participants through three basic steps: reflective engagement with a text or patient narrative, a writing exercise, and sharing or responding within a group setting [32]. The third step of sharing and group discussion serves to solidify concepts and foster community. In a medical student perspective, it was felt that this stage allows for development of communication skills, generates closeness with participants, and promotes building empathy for both patients and peers [33]. Moving forward, it seems that in structuring narrative medicine curricula, the inclusion of a sharing and response component is a beneficial supplement.

In palliative care training, narrative medicine has great potential for enhancing skillsets critical to the field. Our next step may be to consider narrative as a component of fellowship didactics or continuing education to develop these qualities for palliative care providers in training. To do this, programs could consider reading non-fiction texts followed by group discussion, add reflective writing exercises, or encourage other forms of creative endeavors. Fortunately within palliative care, we already have a model for inclusion of multiple perspectives and schools of training. Substantial research shows that palliative care benefits from the interprofessional model of care, including faculty and trainees from domains such as social work, chaplaincy, psychology, child life, nursing, physicians, among others, allowing individuals from various backgrounds and perspectives to come together to best serve the patient [34]. Adding literary scholars and artists to the educational team would likely not feel as so much of a leap within this field. To maintain cohesion within a diverse group does require work, but narrative may also be a tool to provide such structure. One example has been seen among an interprofessional palliative care team where a brief group reflection or “Thought of the Day” after engagement with a text was felt to be a meaningful use of time, and a chance to encourage teamwork [35]. This, too, could be used among interprofessional trainees to support multiple viewpoints and enhance team dynamics.

In pediatrics, the use of narrative may need to be modified, but is still helpful in developing relationships with patients and families. As pediatricians, we are accustomed to searching for narratives among the silences. This occurs across the developmental spectrum: in infants who are unable to utilize language for communication, and in developmentally mature adolescents who may not be able to effectively articulate due to “social voicelessness” [36]. When studying narrative, attention to texts that highlight form in addition to function may be most useful. With its natural emphasis on structure, pattern, and silence, poetry can be a valuable tool to teach providers to identify emotive subtleties, and has been found to be effective in teaching narrative competence among pediatric trainees [37]. Targeting training to emphasize certain skillsets can help to promote competence in identifying patients’ fears and needs, even when they cannot speak for themselves.

For PPC training in particular, there are many desirable skills that narrative medicine can help to build. For children living with a life-threatening condition, care most often includes integration of curative approach with palliation, with some fluidity between the domains [38]. To do this requires the palliative care team to tolerate uncertainty and develop strong communication, skills that, as previously discussed, can be honed with narrative training. While we can try to extrapolate to see where narrative medicine may benefit PPC trainees and patients, it seems that additional research would be helpful to fully understand where narrative medicine may be best utilized. Although there are few studies that look at the use of narrative with patients and caregivers, there is some evidence that suggests utilizing writing may help to promote resiliency within these groups [39]. In PPC where providers look to offer support for siblings, caregivers, and even communities, it seems that writing may be one technique to teach reflection and provide a safe space to process grief for families. Other areas for potential further research could include reading children’s literature with patients to encourage more open discussion, and using narrative practices specifically for bereavement among both caregivers and providers.

It is only reasonable that with the addition of a new curriculum or change in practice clinicians may encounter certain obstacles. One of the most common concerns bound to arise in implementing such a program may be perceived lack of time. Shorter more frequent practices may be one answer, such as implementing a daily introduction of a short poem, piece of music, visual art, or prose, and asking groups to reflect briefly through written or verbal commentary on their experience of interacting with the piece. Other shorter practices include writing a six word essay, or providing 10 min to articulate feelings or inspiration for the day. These activities are brief, and may serve to importantly reframe the day’s work. Longer activities could replace some aspects of traditional didactic curricula for trainees, may be included in retreat workshops, and can be implemented with support to provide continuing medical education (CME) credit.

## 8. Conclusions

In caring for those with serious illness, we may feel helpless entering the room with no cure to offer. As I stood at the threshold of baby R’s room, looking back between his limp form and the devastating MRI, I felt like I was out of options. To me, feeling empty-handed in the face of suffering was one of my greatest fears early on in palliative care fellowship. I learned quickly through mentorship and experience that just my presence and ability to listen carefully in and of itself can be palliative. Sometimes taking the time to bear witness to hardship, absorbing suffering with a nonjudgmental mind, and maintaining empathy in the face of flooding grief, fatigue, and occasionally anger is just what we have to offer as PPC physicians, and is just what is needed. Occasionally the stories are silent, as I witness the ocean of pain shared through a heaving chest. At times they are loud, passionate, yelled from the bedside and riddled with the hunger for someone to listen and stay.

As a new member of this field one of the questions I have heard the most this year is “that seems hard, how do you do it?” My primary drive is the love of stories, and uncovering the story with each encounter. I believe to do the work of PPC and to avoid falling into the claws of burnout, we must find new curiosity and compassion with each clinical encounter, and we must humanize our patients by listening generously. When I reflect on baby R, I do not recall him as the 6-month-old male with meningitis, but rather, as a beloved child with a nuanced story. In writing about baby R, I inadvertently discovered compassion for myself, and after a brief letter correspondence with his parents, I learned it may have been helpful for their bereavement too. Moreover, reading literature and engaging with others who appreciate sharing in fiction has been invaluable in my medical training. I consistently see the bridges between literary works and good medical care, and I hope to integrate narrative deeper into medical education. As palliative care physicians, we deserve to carve out time for sustainable self-care practice and building empathy, and I argue that our trainees’ education also demands it. To truly serve our patients and care for ourselves, our next step is to read, write, and study stories.

## Figures and Tables

**Figure 1 children-08-00321-f001:**
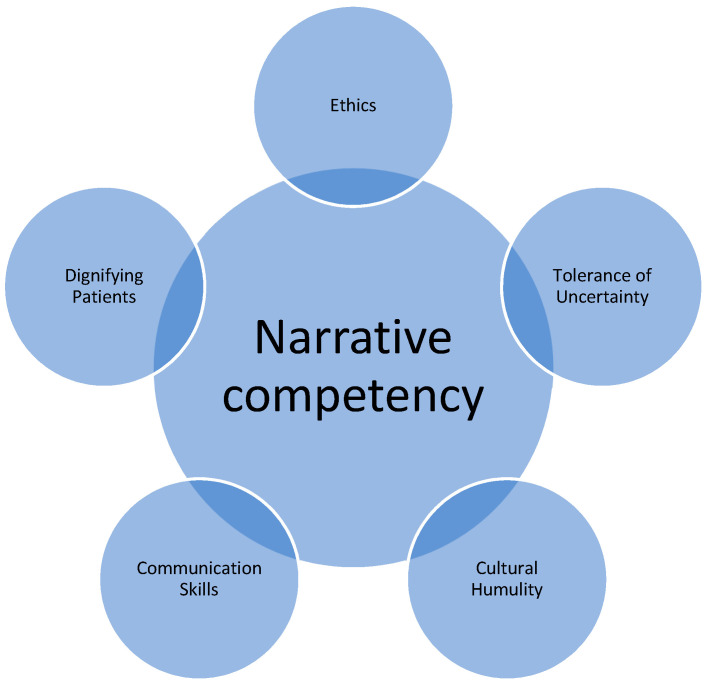
Domains bolstered by narrative competency.

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
