# Peer review of "Lessons in Stories: Why Narrative Medicine Has a Role in Pediatric Palliative Care Training"

_children, 2021, doi:10.3390/children8050321_

Round 1
Reviewer 1 Report
I appreciate the opportunity to review this manuscript and hope my comments assist in the revision process. The material is interesting, the topic is relevant and the author was well-positioned to conduct the opinion article. Despite these positives, in my view, the paper needs more work before it could be published and I have made some specific suggestions below.
Opinion articles that give the authors’ perspective on a topical issue and make a useful addition to the scientific literature. Where appropriate, the author should provide a balanced view of different opinions in the field, and make it clear where they are expressing their own personal views and why. Opinion pieces are also relatively short articles. I think the text would benefit from a greater capacity for synthesis on the part of the author. Brief synopsis or syntheses of ideas and relationship between or within constructs would improve flow dramatically.
- The abstract in question would benefit from some form of framing of the context of the study, by which I mean the history of the problem and the results already formulated that are relevant to it. This would allow a better understanding of the importance of the topic.
- Please provide an introduction to the topic, outlining existing opinions and/or models in the field, and ensure that your opinions and any statements are backed up by supporting information and references, as appropriate.
- A restructuring of the writing to provide more coherent and connected ideas and sections would be valuable.
Reviewer 2 Report
I have reviewed the revised manuscript and feel that my previous recommendations were well addressed. I do not have any further comments for the author. My decision is accept as is. Thanks
Author Response
Thank you for the previous feedback, which was helpful in this round of edits. I appreciate your thoughtful review and decision to accept.
Round 2
Reviewer 1 Report
I appreciate the changes the authors have made, and find the paper now more readable. Relatively to the structure of the article, and in compliance with the guidelines of the editor (perspective piece), it seems to me to fulfil the purpose for which it is intended.
- The English is generally good, but occasionally with mistakes in grammar and syntax and with prepositions and other words that seem to be wrong. I would recommend that the text is reviewed by a linguistic professional.
This manuscript is a resubmission of an earlier submission. The following is a list of the peer review reports and author responses from that submission.
Round 1
Reviewer 1 Report
Thank you for allowing me the opportunity to review “Lessons in Stories: Why Narrative Medicine has a Role in Pediatric Palliative Care Training.” This manuscript is vastly different than what is usually published in scientific journals in several important ways: it is written in the first person perspective, it briefly reviews literature (and does not do so in a systematic way), and presents no new quantitative data. Yet, the manuscript is beautifully written and has a message that I think is of great scientific value – not only to pediatric palliative care doctors, but to all pediatric physicians in general (and potentially to all physicians). There are suggestions that in my opinion would further strengthen the paper. First, it would be nice if the author would add a paragraph about concrete “how to.” For example, if a physician is interested in implementing narrative into their clinical work/practice, how might they go about doing that? Aside from reading and writing, are there training resources for how to incorporate narrative into their work? If so, they should be listed. If not, then it would be nice for the author to provide this guidance. Also, how might a program go about adding this to their curriculum? Are there universities that already do this? If so, describing those extant models would be helpful. Second, it would be nice for the author to provide a more thorough review of the literature of narrative in medical setting. Is there evidence for this reducing burnout? Is there evidence on how it impacts patient satisfaction, etc? Finally, it would be nice to have a paragraph of overcoming obstacles/concerns. For example, how would you respond to the common concern that “I don’t have time for that”? Discussing the commonly-encountered points of resistance to incorporating narrative would significantly strengthen the paper.
Reviewer 2 Report
This is an important study in an under-researched area of the world. However it needs a considerable amount of work. Some areas need clarification as noted below:
- Situate the concept of narrative medicine within the context of extant medical/palliative care knowledge. Discuss the international relevance of the concept and describe thoroughly the rationale for the review in the context of what is already known. More actual references should be provided.
The article as it is constructed looks more like a reflection than a review. The main and fundamental purpose of writing a review is to create a readable synthesis of the best resources available in the literature for an important research question or a current area of research. Given the approach to a narrative/integrative review it seems to me important to provide an explicit statement of questions being addressed with reference to participants, interventions, comparisons, outcomes (PICO acronym).
- Identify the data bases searched, with inclusive dates of the literature searched for each database and keywords used. Do not include when the literature actually was searched.
- Discuss retrieval of references and handling, including inclusion and exclusion criteria (i.e., how the analysis was conducted, including judgment of quality of papers included in the literature review). I suggest a better explanation of the criteria for inclusion/exclusion (e.g. PICO method) used in the selection of the studies analyzed. Regarding the high number of initial referrals, it would be important to define whether this sample resulted in a basic research/ individual from each descriptor or is an advanced search and resulted therefore the intersection of different descriptors (e.g. Boolean method). Present full electronic search strategy, such that it could be repeated.
- Describe method of data extraction from reports (e.g., piloted forms, independently, in duplicate) and any processes for obtaining and confirming data.
- Results of the study should be discussed relative to the literature; however a quantitative examination would be interesting to integrate, and not only a qualitative analysis.
Discussion
- Please introduce study limitations. The limitations of the study could easily be addressed and incorporated within the discussion of this section (one important limitation it is probably the heterogeneity of primary studies).
- Implications for paediatric palliative practice and research need addressed in more deep. Identify implications/ recommendations for healthcare providers, educators, and policy makers as appropriate, and consistent with limitations.
CHECKLIST FOR STYLE
Organization and style: The manuscript is clearly written and will serve a broad audience of students, researchers, and practitioners.
Reviewer 3 Report
I agree with the author that competencies in narrative medicine have a key role, not only in PPC, but for all students and professionals of Medicine.